# A Deep Learning Model for Prostate Adenocarcinoma Classification in Needle Biopsy Whole-Slide Images Using Transfer Learning

**DOI:** 10.3390/diagnostics12030768

**Published:** 2022-03-21

**Authors:** Masayuki Tsuneki, Makoto Abe, Fahdi Kanavati

**Affiliations:** 1Medmain Research, Medmain Inc., Fukuoka 810-0042, Fukuoka, Japan; fkanavati@medmain.com; 2Department of Pathology, Tochigi Cancer Center, 4-9-13 Yohnan, Utsunomiya 320-0834, Tochigi, Japan; makotabe@tochigi-cc.jp

**Keywords:** deep learning, adenocarcinoma, prostate, biopsy, whole-slide image, transfer learning

## Abstract

The histopathological diagnosis of prostate adenocarcinoma in needle biopsy specimens is of pivotal importance for determining optimum prostate cancer treatment. Since diagnosing a large number of cases containing 12 core biopsy specimens by pathologists using a microscope is time-consuming manual system and limited in terms of human resources, it is necessary to develop new techniques that can rapidly and accurately screen large numbers of histopathological prostate needle biopsy specimens. Computational pathology applications that can assist pathologists in detecting and classifying prostate adenocarcinoma from whole-slide images (WSIs) would be of great benefit for routine pathological practice. In this paper, we trained deep learning models capable of classifying needle biopsy WSIs into adenocarcinoma and benign (non-neoplastic) lesions. We evaluated the models on needle biopsy, transurethral resection of the prostate (TUR-P), and The Cancer Genome Atlas (TCGA) public dataset test sets, achieving an ROC-AUC up to 0.978 in needle biopsy test sets and up to 0.9873 in TCGA test sets for adenocarcinoma.

## 1. Introduction

According to the Global Cancer Statistics 2020, prostate cancer was the second-most-frequent cancer and the fifth leading cause of cancer death among men in 2020 with an estimated 1,414,259 new cases and 375,304 deaths worldwide, which is the most frequently diagnosed cancer in men in over one-half (112 of 185) of the countries [1].

Serum prostate-specific antigen (PSA) is the most important and clinically useful biochemical marker in prostate [2]. PSA has contributed to an increase in the early detection rate of prostate cancer and is now advocated for routine use for screening in men [2]. Serum PSA is also an important tool in the management of prostate cancer. Elevation of PSA correlates with cancer recurrence and progression after treatment. Thus, PSA is a sensitive marker for tumor recurrence after treatment and is useful for the early detection of metastases. However, n elevated serum PSA concentration is seen not only in patients with adenocarcinoma, but also in patients with aging, prostatitis, benign prostatic hyperplasia, and transiently following biopsy [3,4,5]. Although PSA elevations might indicate the presence of prostate disease (e.g., prostate cancer, benign prostatic hyperplasia, and prostatitis), not all men with prostate disease have elevated PSA levels, and PSA elevations are not specific for prostate cancer. Therefore, it is necessary to perform definitive diagnosis of the presence of prostate adenocarcinoma by needle biopsy for cancer treatment.

As for needle biopsy, in the past, the standard approach was to take six cores (sextant biopsies) [6]. However, based on a systematic review [7], it has been shown that cancer yield was significantly associated with increasing number of cores, more so in the case of laterally directed cores than centrally directed cores. This is based on the finding that schemes with 12 laterally directed cores detected 31% more cancers than the six cores. Schemes with further cores (18 to 24) showed no further gains in cancer detection. Hence, a 12-core systematic biopsy that incorporates apical and far-lateral cores in the template distribution allows maximal cancer detection, avoids repeat biopsy, and provides information adequate for identifying men who need cancer treatment [8]. However, diagnosing a large number of cases containing 12 core biopsy specimens is a time-consuming manual system for pathologists in routine practice.

Adenocarcinoma is by far the most common malignant tumor of the prostate gland. Adenocarcinoma tends to be multifocal with a predilection for the peripheral zone. Histopathologically, the majority of prostate adenocarcinomas are not difficult to diagnose. However, the separation of well-differentiated adenocarcinoma from the vast number of benign prostatic hyperplasia or atypical gland proliferation, the detection of small adenocarcinoma foci, and the differentiation of poorly differentiated adenocarcinoma from inflammatory cell infiltration are sometimes very challenging in routine diagnoses.

Therefore, all these factors mentioned above highlight the benefit of establishing a histopathological screening system based on needle biopsy specimens for prostate adenocarcinoma patients. Conventional morphological diagnosis by human pathologists has limitations, and it is necessary to construct a new diagnostic strategy based on the analysis of a large number of cases in the future.

Deep learning has been widely applied in computational histopathology, with applications such as cancer classification in whole-slide images (WSIs), cell detection and segmentation, and the stratification of patient outcomes [9,10,11,12,13,14,15,16,17,18,19,20,21,22]. For prostate histopathology in particular, deep learning has been applied for the classification of cancer in WSIs [21,23,24,25,26,27,28,29,30].

In this study, we trained a WSI prostate adenocarcinoma classification model using transfer learning and weakly supervised learning. We evaluated the models on needle biopsy, transurethral resection of the prostate (TUR-P), and TCGA public dataset test sets to confirm application of the algorithm in different types of specimens, achieving an ROC-AUC up to 0.978 in needle biopsy test sets and up to 0.9873 in The Cancer Genome Atlas (TCGA) test sets for adenocarcinoma. We also evaluated on the needle biopsy test sets, without fine-tuning, models that had been previously trained on other organs for the classification of adenocarcinomas [22,31,32,33,34,35,36,37]. These findings suggest that computational algorithms might be useful as routine histopathological diagnostic aids for prostate adenocarcinoma classification.

## 2. Materials and Methods

### 2.1. Clinical Cases and Pathological Records

This was a retrospective study. A total of 2926 hematoxylin and eosin (H&E)-stained histopathological specimens of human prostate adenocarcinoma and benign lesions—1682 needle biopsy and 1244 TUR-P—were collected from the surgical pathology files of five hospitals: Shinyukuhashi, Wajiro, Shinkuki, Shinkomonji, and Shinmizumaki hospitals (Kamachi Group Hospitals, Fukuoka, Japan), after histopathological review of those specimens by surgical pathologists. The cases were selected randomly so as to reflect a real clinical scenario as much as possible. The pathologists excluded cases that had poor scan quality. Each WSI diagnosis was observed by at least two pathologists, with the final checking and verification performed by a senior pathologist. All WSIs were scanned at a magnification of 20× using the same Leica Aperio AT2 scanner and were saved in the SVS file format with JPEG2000 compression.

### 2.2. Dataset

Table 1 and Table 2 break down the distribution of the dataset into training, validation, and test sets. The training and validation sets consisted of needle biopsy WSIs (Table 1). The test sets consisted of needle biopsy, TUR-P, and TCGA public dataset WSIs (Table 2). The regions of the prostate sampled by TUR-P and needle biopsy tend to be different. TUR-P specimens usually consist of tissues from the transition zone, urethra, periurethral area, bladder neck, anterior fibromuscular stroma, and occasionally, small portions of seminal vesicles. In contrast, most needle biopsy specimens consist mainly of tissue from the peripheral zone. The split was carried out randomly taking into account the proportion of each label in the dataset. Hospitals that provided histopathological cases were anonymized (e.g., Hospital A, Hospital B). The patients’ pathological records were used to extract the WSIs’ pathological diagnoses and to assign WSI labels. Training set WSIs were not annotated, and the training algorithm only used the WSI diagnosis labels, meaning that the only information available for the training was whether the WSI contained adenocarcinoma or was benign (non-neoplastic), but no information about the location of the cancerous tissue lesions.

### 2.3. Deep Learning Models

We trained the models using the partial fine-tuning approach [38]. It consisted of using the weights of an existing pre-trained model and only fine-tuning the affine parameters of the batch normalization layers and the final classification layer. We used the EfficientNetB1 [39] model starting with pre-trained weights on ImageNet. Figure 1 shows an overview of the training method.

The training method that we used in this study was exactly the same as reported in a previous study [34]. For completeness, we repeat the method here. To apply the CNN on the WSIs, we performed slide tiling by extracting square tiles from tissue regions. On a given WSI, we detected the tissue regions and eliminated most of the white background by performing a thresholding on a grayscale version of the WSI using Otsu’s method [40]. During prediction, we performed the tiling in a sliding window fashion, using a fixed-size stride, to obtain predictions for all the tissue regions. During training, we initially performed random balanced sampling of tiles from the tissue regions, where we tried to maintain an equal balance of each label in the training batch. To do so, we placed the WSIs in a shuffled queue such that we looped over the labels in succession (i.e., we alternated between picking a WSI with a positive label and a negative label). Once a WSI was selected, we randomly sampled $\frac{\mathrm{batch}\;\mathrm{size}}{\mathrm{num}\;\mathrm{labels}}$ tiles from each WSI to form a balanced batch. To maintain the balance on the WSI, we oversampled from the WSIs to ensure the model trained on tiles from all of the WSIs in each epoch. We then switched to the hard mining of tiles once there was no longer any improvement on the validation set after two epochs. To perform the hard mining, we alternated between training and inference. During inference, the CNN was applied in a sliding window fashion on all of the tissue regions in the WSI, and we then selected the *k* tiles with the highest probability for being positive if the WSI was negative and the *k* tiles with the lowest probability for being positive if the WSI was positive. This step effectively selected the hard examples with which the model was struggling. The selected tiles were placed in a training subset, and once that subset contained *N* tiles, the training was run. We used k=8, N=256, and a batch size of 32.

To obtain a prediction on a WSI, the model was applied in a sliding window fashion, generating a prediction per tile. The WSI prediction was then obtained by taking the maximum from all of the tiles.

We trained the models with the Adam optimization algorithm [41] with the following parameters: $beta_{1}=0.9$, $beta_{2}=0.999$, and a batch size of 32. We used a learning rate of 0.001 when fine-tuning. We applied a learning rate decay of 0.95 every 2 epochs. We used the binary cross-entropy loss function. We used early stopping by tracking the performance of the model on a validation set, and training was stopped automatically when there was no further improvement on the validation loss for 10 epochs. The model with the lowest validation loss was chosen as the final model.

### 2.4. Software and Statistical Analysis

The deep learning models were implemented and trained using TensorFlow [42]. AUCs were calculated in Python using the scikit-learn package [43] and plotted using matplotlib [44]. The 95% CIs of the AUCs were estimated using the bootstrap method [45] with 1000 iterations.

The true positive rate (TPR) was computed as:(1)$$TPR=\frac{TP}{TP+FN}$$
and the false positive rate (FPR) was computed as:(2)$$FPR=\frac{FP}{FP+TN}$$
where TP, FP, and TN represent true positive, false positive, and true negative, respectively. The ROC curve was computed by varying the probability threshold from 0.0 to 1.0 and computing both the TPR and FPR at the given threshold.

### 2.5. Code Availability

To train the classification model in this study, we used the publicly available TensorFlow training script available at https://github.com/tensorflow/models/tree/master/official/vision/image_classification, accessed on 23 March 2021.

## 3. Results

### 3.1. High AUC Performance of the WSI Evaluation of Prostate Adenocarcinoma Histopathology Images in the Needle Biopsy, TUR-P, and TCGA Test Sets

The aim of this retrospective study was to train deep learning models for the classification of prostate adenocarcinoma in WSIs of prostate needle biopsy specimens. We had a total of 1122 needle biopsy WSIs (438 adenocarcinoma and 684 benign WSIs) for the training set and a total of 60 WSIs (30 adenocarcinoma and 30 benign WSIs) for the validation set from five sources (Hospitals A, B, C, D, and E) (Table 1). We used a transfer learning (TL) approach based on partial fine-tuning [38] to train the models. We refer to the trained models as TL <magnification> <tile size> <model size>, based on the different configurations. As we had at our disposal ten models that had been trained specifically on specimens from different organs (breast, colon, stomach, pancreas, and lung) [22,31,32,33,34,35,36,37], we evaluated these models without fine-tuning on the biopsy test sets (Hospitals A–C) (Table 2) to investigate whether morphological cancer similarities transferred across organs without additional training. Table 3 breaks down the values of ROC-AUC and log loss in the biopsy test set (Hospitals A–C) and shows that the colon poorly differentiated adenocarcinoma model (colon poorly ADC-2 (20×, 512)) [36] exhibited the highest ROC-AUC (0.8172, CI: 0.7815–0.855) and the lowest log loss (0.5216, CI: 0.4748–0.5695), indicating its capability as a base model for the transfer learning approach.

Overall, we trained three different models: (1) a transfer learning model (TL-colon poorly ADC-2 (20×, 512)) using the existing colon poorly differentiated adenocarcinoma model (colon poorly ADC-2 (20×, 512)) [36] at a magnification 20× and a tile size of 512 px × 512 px; (2) a model (EfficientNetB1 (20×, 512)) using the EfficientNetB1 at magnification 20× and a tile size of 512 px × 512 px, starting with pre-trained weights from ImageNet; (3) a model (EfficientNetB1 (10×, 224)) using the EfficientNetB1 at magnification 10× and a tile size of 224 px × 224 px, starting with pre-trained weights from ImageNet.

We evaluated the trained models on the needle biopsy, TUR-P, and TCGA test sets (Table 2). We confirmed that the surgical pathologists were able to diagnose these cases from visual inspection of the H&E-stained slides alone prior to the test sets’ evaluation. The distribution of the number of WSIs in each test set is summarized in Table 2. For each test set, we computed the ROC-AUC, log loss, accuracy, sensitivity, and specificity, and we summarize the results in Table 4 and Table 5 and Figure 2. In Table 4, we compare the results of the ROC-AUC and log loss among three models (TL-colon poorly ADC-2 (20×, 512), EfficientNetB1 (20×, 512), and EfficientNetB1 (10×, 224)) we trained.

The model (TL-colon poorly ADC-2 (20×, 512)) achieved the highest ROC-AUCs of 0.9873 (CI: 0.9881-0.995) and the lowest log loss of 0.0742 (CI: 0.0551–0.0959) for prostate adenocarcinoma on the TCGA test set (Table 4). On the needle biopsy test set, the model (TL-colon poorly ADC-2 (20×, 512)) also achieved very high ROC-AUCs (0.967–0.978) with low values of the log loss (0.2094–0.3788) (Table 4). In contrast, ROC-AUCs on the TUR-P test set were lower than the biopsy test set, and the log loss on the TUR-P test set was higher than the biopsy test set (Table 4). In addition, accuracy, sensitivity, and specificity results on the model (TL-colon poorly ADC-2 (20×, 512)) on the biopsy, TUR-P, and TCGA test sets are given in Table 5. The model (TL-colon poorly ADC-2 (20×, 512)) achieved very high accuracy (0.918–0.949), sensitivity (0.89–0.948), and specificity (0.924–0.98) on the biopsy and TCGA test sets (Table 5). On the TUR-P test sets, the model (TL-colon poorly ADC-2 (20×, 512)) achieved high accuracy (0.8902–0.9176) and specificity (0.9247–0.9545), but low sensitivity (0.4151–0.7982) (Table 5). As shown in Figure 2, the model (TL-colon poorly ADC-2 (20×, 512)) is fully applicable for prostate adenocarcinoma classification on the needle biopsy WSIs, as well as the TCGA public WSI dataset, but not on the TUR-P WSIs.

Figure 3, Figure 4, Figure 5, Figure 6 and Figure 7 show representative cases of true positives (biopsy and TUR-P), false positives (biopsy and TUR-P), and false negatives (biopsy), respectively, using the model (TL-colon poorly ADC-2 (20×, 512)).

### 3.2. True Positive Prediction on Needle Biopsy Specimens

Our model (TL-colon poorly ADC-2 (20×, 512)) satisfactorily predicted prostate adenocarcinoma on needle biopsy specimens (Figure 3A). According to the pathological diagnostic report, there were adenocarcinoma foci in two of six needle biopsy cores (#5 and #6), which the pathologists marked as red ink dots (yellow triangles) on the glass slides. The heat map image shows true positive predictions (Figure 3B,D,F,H) of adenocarcinoma cell infiltrating areas (Figure 3C,E,G). In Figure 3G, the pathologists did not mark when they performed the diagnosis; however, the heat map image show true positive predictions of adenocarcinoma foci, which were reviewed and verified as adenocarcinoma by other pathologists (Figure 3H). In contrast, the heat map image does not show true positive predictions on glomeruloid glands precisely, which were assigned a Gleason Pattern 4 [46,47] (Figure 3G,H). Importantly, the heat map images also exhibit a perfect true negative prediction of needle biopsy cores (#1–#4) on the same WSI (Figure 3B).

### 3.3. False Positive Prediction on Needle Biopsy Specimens

Inflammatory tissues (Figure 4A) and prostatic hyperplasia (Figure 4E) were false positively predicted for prostate adenocarcinoma (Figure 4B,F) using the transfer learning model (TL-colon poorly ADC-2 (20×, 512)). In the inflammatory tissue (Figure 4A), the infiltration of chronic inflammatory cells including histiocytes, lymphocytes, and plasma cells (Figure 4C) was the primary cause of the false positive prediction (Figure 4D) due to a morphology analogous to adenocarcinoma cells. Prostatic hyperplasia (Figure 4E) with irregularly shaped and diverse sizes of tubular structures (Figure 4G) was the primary cause of the false positive prediction (Figure 4H).

### 3.4. False Negative Prediction on the Needle Biopsy Specimens

In a representative false negative case (Figure 5A), histopathologically, there were adenocarcinoma foci (Figure 5C–E) in three out of four needle biopsy specimens, which the pathologists marked with blue dots when they performed the pathological diagnoses. However, the heat map image exhibits no true positive predictions (Figure 5B).

### 3.5. True Positive Prediction on the TUR-P Specimens

Although not as accurate as the biopsy specimens (Table 4), there were many cases in which prostate adenocarcinoma could be classified precisely on the TUR-P specimens. In a representative true positive TUR-P case (Figure 6A), the transfer learning model (TL-colon poorly ADC-2 (20×, 512)) satisfactorily predicted prostate the adenocarcinoma-invading area (Figure 6B). The heat map image shows the true positive predictions of adenocarcinoma cell infiltration (Figure 6C,D) with the true negative prediction of prostatic hyperplasia (Figure 6A,B).

### 3.6. False Positive Prediction on TUR-P Specimens

By the transfer learning model (TL-colon poorly ADC-2 (20×, 512)), false positives on the TUR-P specimens were not only due to prostatic hyperplasia, as observed for the needle biopsy specimens (Figure 4E–H), but also due to inflammation (Figure 7A–D) and false positives coinciding with areas of tissue degeneration caused by thermal ablation at the specimen margins (Figure 7E–H) because in TUR-P, the endoscope is inserted into the prostate through the urethra and the tissue is harvested with an electrocautery, resulting in marginal degeneration of the specimen due to thermal cauterization.

## 4. Discussion

In this study, we trained deep learning models for the classification of prostate adenocarcinoma in needle biopsy WSIs. Of the three models we trained (Table 4), the best model (TL-colon poorly ADC-2 (20×, 512)) achieved ROC-AUCs in the range of 0.967–0.978 on the needle biopsy, in the range of 0.7377–0.9098 on the TUR-P, and 0.9873 on the TCGA public datasets. The other two models were trained using the EfficientNetB1 [39] model starting with pre-rained weights on ImageNet at different magnifications (10×, 20×) and tile sizes (224 × 224, 512 × 512). The model based on EfficientNetB1 (EfficientNetB1 (20×, 512)) achieved high ROC-AUCs in close proximity to the values of, but lower than, the best model (TL-colon poorly ADC-2 (20×, 512)). The best model (TL-colon poorly ADC-2 (20×, 512)) was trained by the transfer learning approach based on our existing colon poorly differentiated adenocarcinoma classification model [36]. To train the models, we used only 1122 needle biopsy WSIs (adenocarcinoma: 438 WSIs, benign: 684 WSIs) without manual annotations by the pathologists [22,37], as compared to the previous study (about 8400 needle biopsy WSIs for training) [21]. However, we needed to train the models for TUR-P WSIs separately in the next step because TUR-P WSIs were not applicable to be predicted precisely by the best model (TL-colon poorly ADC-2 (20×, 512)).

The best model (TL-colon poorly ADC-2 (20×, 512)) achieved similar values of the ROC-AUC, log loss, accuracy, sensitivity, and specificity among three independent medical institutes (Hospitals A, B, C) and the TCGA public dataset test sets (Table 4 and Table 5), meaning that the best model has general versatility in prostate needle biopsy WSIs.

Various benign (non-neoplastic) lesions can mimic adenocarcinoma on needle biopsy specimens, which include glandular lesions such as adenosis, atrophy, verumontanum mucosal gland hyperplasia, atypical adenomatous hyperplasia, nephrogenic metaplasia, hyperplasia of mesonephric remnants, and basal cell hyperplasia [48]. Inflammation (acute and chronic or granulomatous prostatitis) and prostatic hyperplasia are often present in needle biopsy specimens, and they may become problematic to differentiate between benign and adenocarcinoma if their histopathological features are similar to adenocarcinoma in routine diagnosis. Similar to human pathologists, the major causes for false positives predicted by the best model (TL-colon poorly ADC-2 (20×, 512)) were inflammatory cell infiltration, especially histiocytes, lymphocytes, and plasma cells, which morphologically mimic adenocarcinoma cells and prostatic hyperplasia with irregularly shaped and different sizes of tubular structures (Figure 4). In addition, normal benign prostate tissues including seminal vesicles, paraganglia, and ganglion cells may also be confused histopathologically with adenocarcinoma in needle biopsy specimens [48], which were also predicted as adenocarcinoma at the tile level in the small areas of false positively predicted WSIs in this study. Moreover, in routine clinical practice, prostate adenocarcinoma with atrophic features is easily confused with benign acinar atrophy [49], which may cause false negative prediction by deep learning models. It may be necessary to add controversial prostate adenocarcinoma and benign WSIs, which are more likely to cause false positives and false negatives, to attempt to further improve the model’s performance on such cases. Interestingly, false positive predictions in cauterized areas of the marginal zone of the specimens were characteristic of TUR-P WSIs (Figure 7). The lower observed results on TUR-P were potentially due to the presence of prostate hyperplasia, which morphologically mimics prostate adenocarcinoma. This indicates that to further improve performance on TUR-P cases, we would require a training set that would specifically account for such cases so as to aid the model in reducing false positives.

A greater number of prostate biopsies (usually 12-core systemic biopsy) are performed currently, and more biopsy cores are submitted to surgical pathology than ever before, resulting in a huge interpretive burden for pathologists. Indeed, many patients undergo biopsy for elevated serum PSA with no other clinical evidence of cancer, resulting in an enormous number of biopsies performed even if numerous diagnostic pitfalls (e.g., fatigue, time-consuming workflow) and mimics of prostate cancer have been described. Thus, the ultimate goal of prostate adenocarcinoma detection, as well as the prediction of the outcome for the individual patient should be augmented by deep-learning-based software applications. The deep learning models established in the present study achieved very high ROC-AUC performances (Figure 2 and Table 4) on prostate needle biopsy WSIs; they offer promising results that indicate they could be beneficial as a screening aid for pathologists prior to observing histopathology on glass slides or WSIs. At the same time, it can be used as a double-check to reduce the risk of missed cancer foci. The major advantage of using an automated tool is that it can systematically handle large amounts of WSIs without potential bias due to the fatigue commonly experienced by pathologists, which could drastically alleviate the heavy clinical burden of practical pathology diagnosis using conventional microscopes. While the results are promising, further clinical validation studies are required in order to further evaluate the robustness of the models in a potential clinical setting before they can actually be used in clinical practice. If such models are deemed viable after rigorous clinical validation, they can transform the future of healthcare and precision oncology.

## Figures and Tables

**Figure 1 diagnostics-12-00768-f001:**
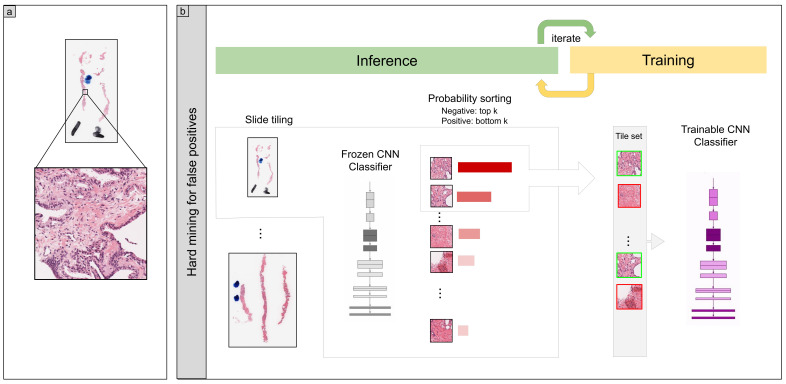
(**a**) shows a zoomed-in example of a tile in a WSI. (**b**) During training, we iteratively alternated between inference and training steps. The model weights were frozen during the inference step, and this was applied in a sliding window fashion on the entire tissue regions of each WSI. The top k tiles with the highest probabilities were then selected from each WSI and placed into a queue. During training, the selected tiles from multiple WSIs formed a training batch and were used to train the model.

**Figure 2 diagnostics-12-00768-f002:**
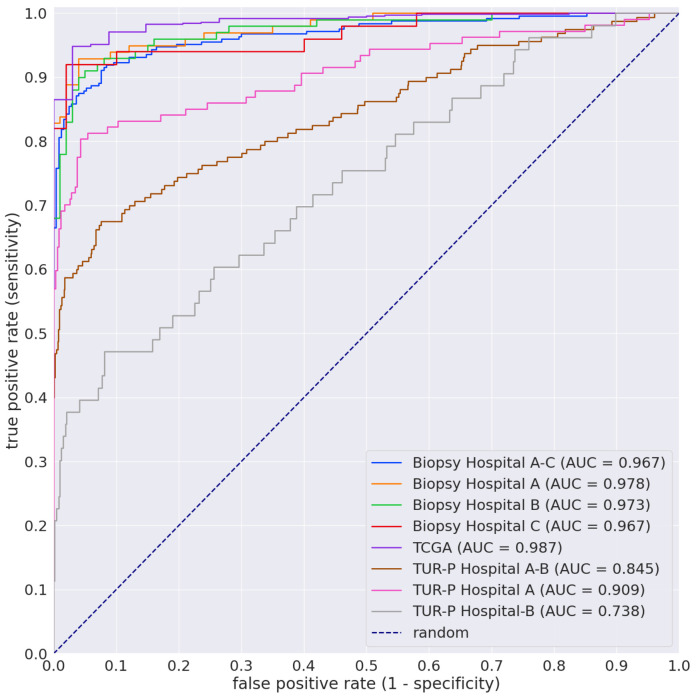
ROC curves on the biopsy (Hospitals A, B, C, and A–C), TUR-P (Hospitals A, B, and A and B), and TCGA test sets of the TL-colon poorly ADC-2 (20×, 512) model.

**Figure 3 diagnostics-12-00768-f003:**
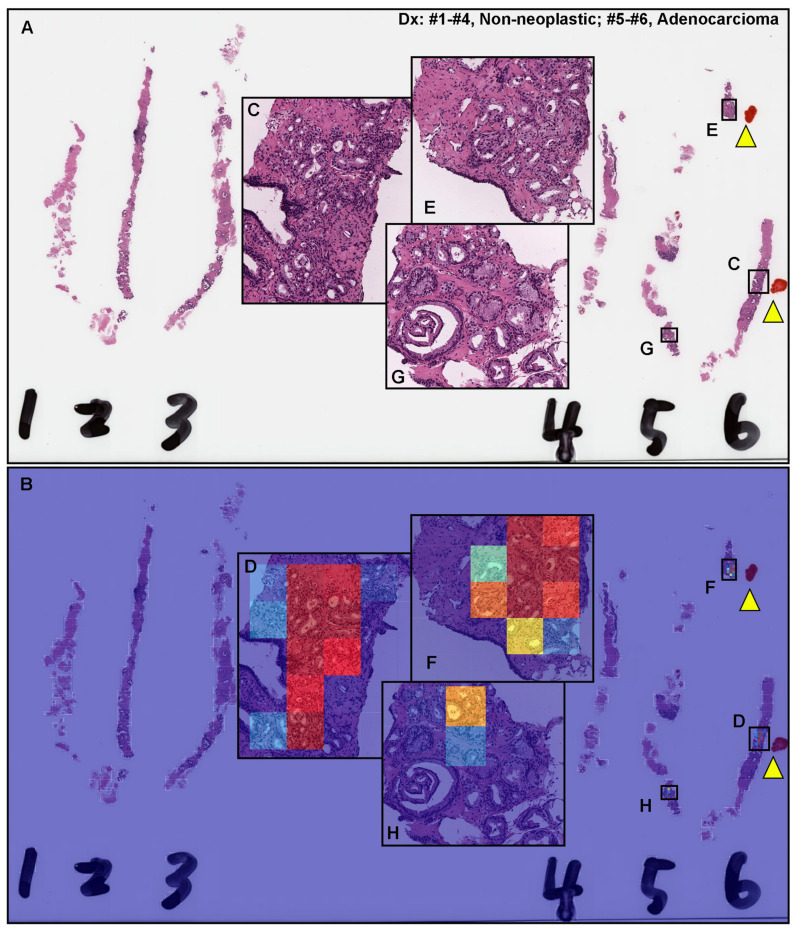
Representative true positive prostate adenocarcinoma from the biopsy test sets. On the prostate needle biopsy whole-slide image (**A**), Specimens #1–#4 are benign (non-neoplastic), and there are adenocarcinoma cell infiltration foci (**C**,**E**,**G**) in Specimens #5 and #6 based on the pathological diagnostic report, which the pathologists marked as red ink dots (yellow triangles) on the glass slides. The heat map image (**B**) shows the true positive prediction of adenocarcinoma cells (**D**,**F**,**H**) using transfer learning from the colon poorly differentiated adenocarcinoma model (TL-colon poorly ADC-2 (20×, 512)), which corresponds respectively to the H&E histopathology (**C**,**E**,**G**). The heat map uses the jet color map where blue indicates low probability and red indicates high probability.

**Figure 4 diagnostics-12-00768-f004:**
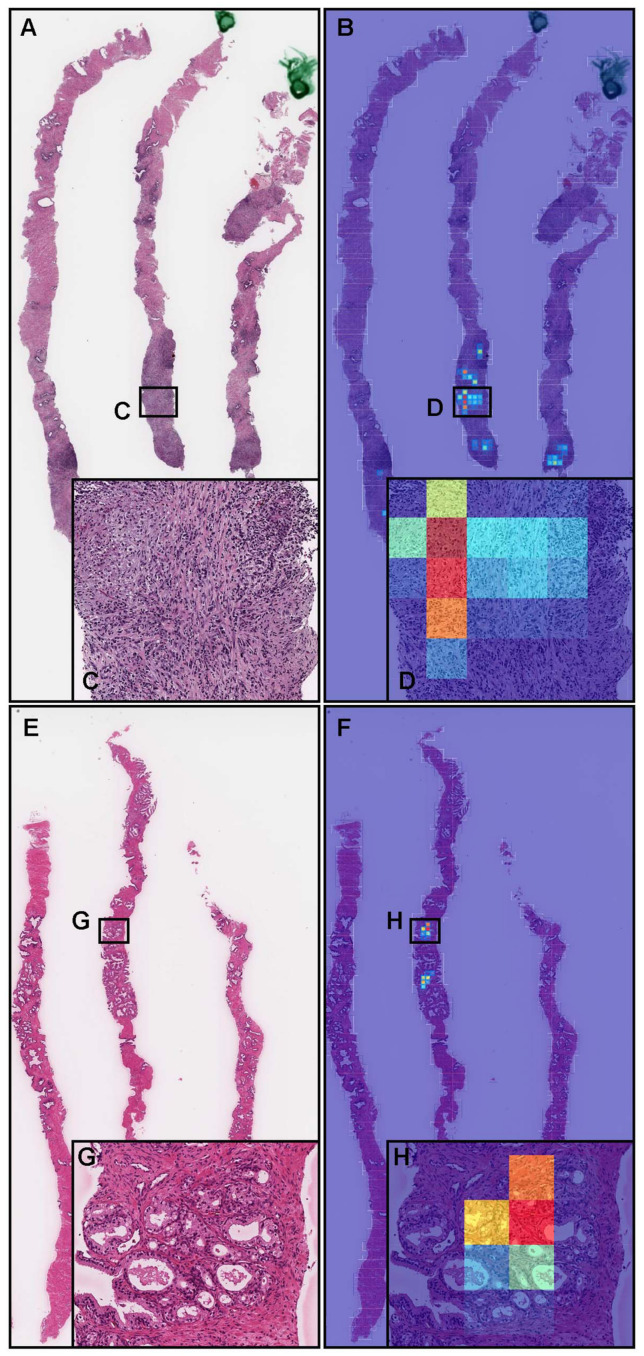
Representative examples of prostate adenocarcinoma false positive prediction outputs on cases from the needle biopsy test sets. Histopathologically, (**A**,**E**) are benign (non-neoplastic) lesions. The heat map images (**B**,**F**) exhibit false positive predictions of adenocarcinoma (**D**,**H**) using transfer learning from the colon poorly differentiated adenocarcinoma model (TL-colon poorly ADC-2 (20×, 512)). Infiltration of chronic inflammatory cells including histiocytes, lymphocytes, and plasma cells (**C**) would be the primary cause of the false positives due to a morphology analogous to adenocarcinoma cells’ infiltration (**D**). Areas where prostatic hyperplasia (**G**) would be the primary cause of false positives (**H**). The heat map uses the jet color map where blue indicates low probability and red indicates high probability.

**Figure 5 diagnostics-12-00768-f005:**
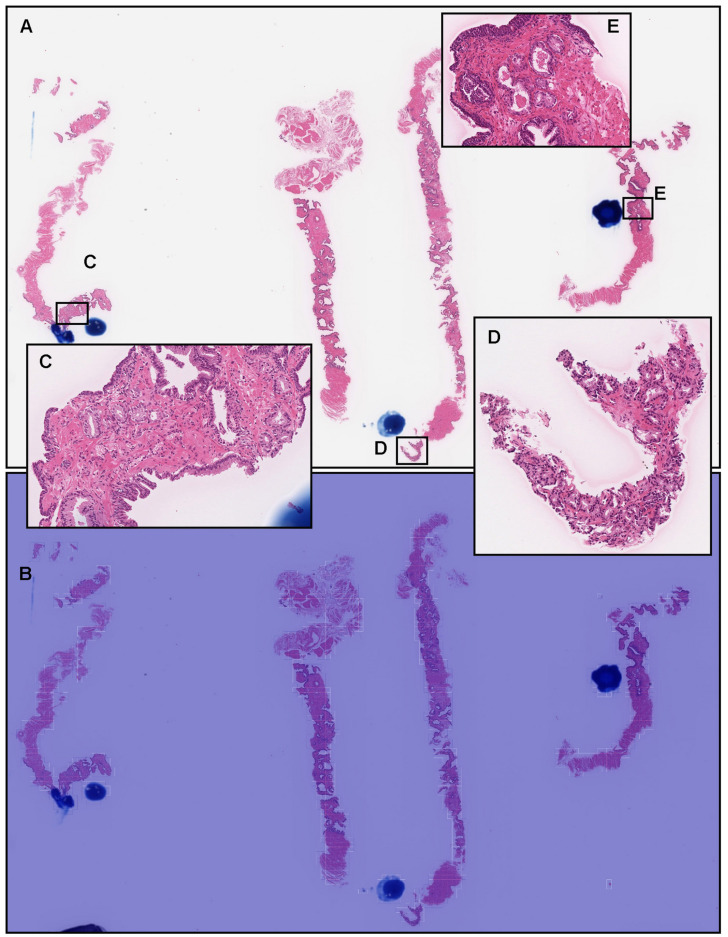
Representative false negative prostate adenocarcinoma from the needle biopsy test sets. According to the histopathological report, there were four needle biopsy specimens in the WSI, and three of them had adenocarcinomas (**A**). The pathologists marked the adenocarcinoma areas in blue dots (**A**). High-power view showing that there were adenocarcinoma foci (**C**–**E**). The heat map image (**B**) shows no true positive predictions of adenocarcinoma using transfer learning from the colon poorly differentiated adenocarcinoma model (TL-colon poorly ADC-2 (20×, 512)).

**Figure 6 diagnostics-12-00768-f006:**
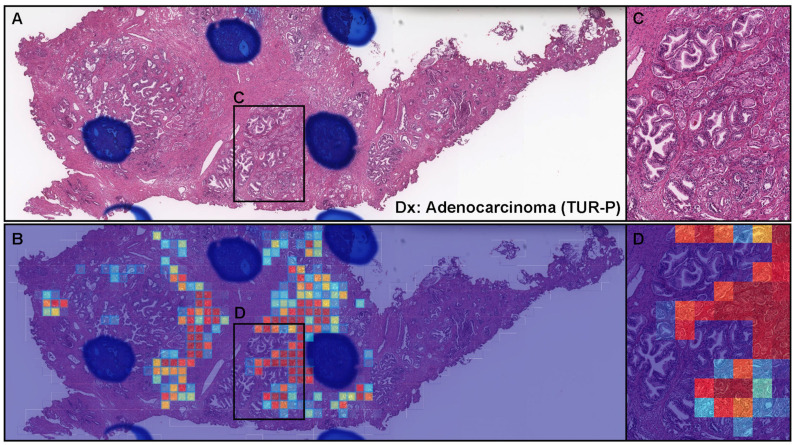
Representative true positive prostate adenocarcinoma from the transurethral resection of the prostate (TUR-P) test sets. In the TUR-P specimen (**A**), there are adenocarcinoma cell infiltration foci (**C**) based on the histopathological report. The heat map image (**B**) shows the true positive prediction of adenocarcinoma cells (**D**) using transfer learning from the colon poorly differentiated adenocarcinoma model (TL-colon poorly ADC-2 (20×, 512)). The heat map uses the jet color map where blue indicates low probability and red indicates high probability.

**Figure 7 diagnostics-12-00768-f007:**
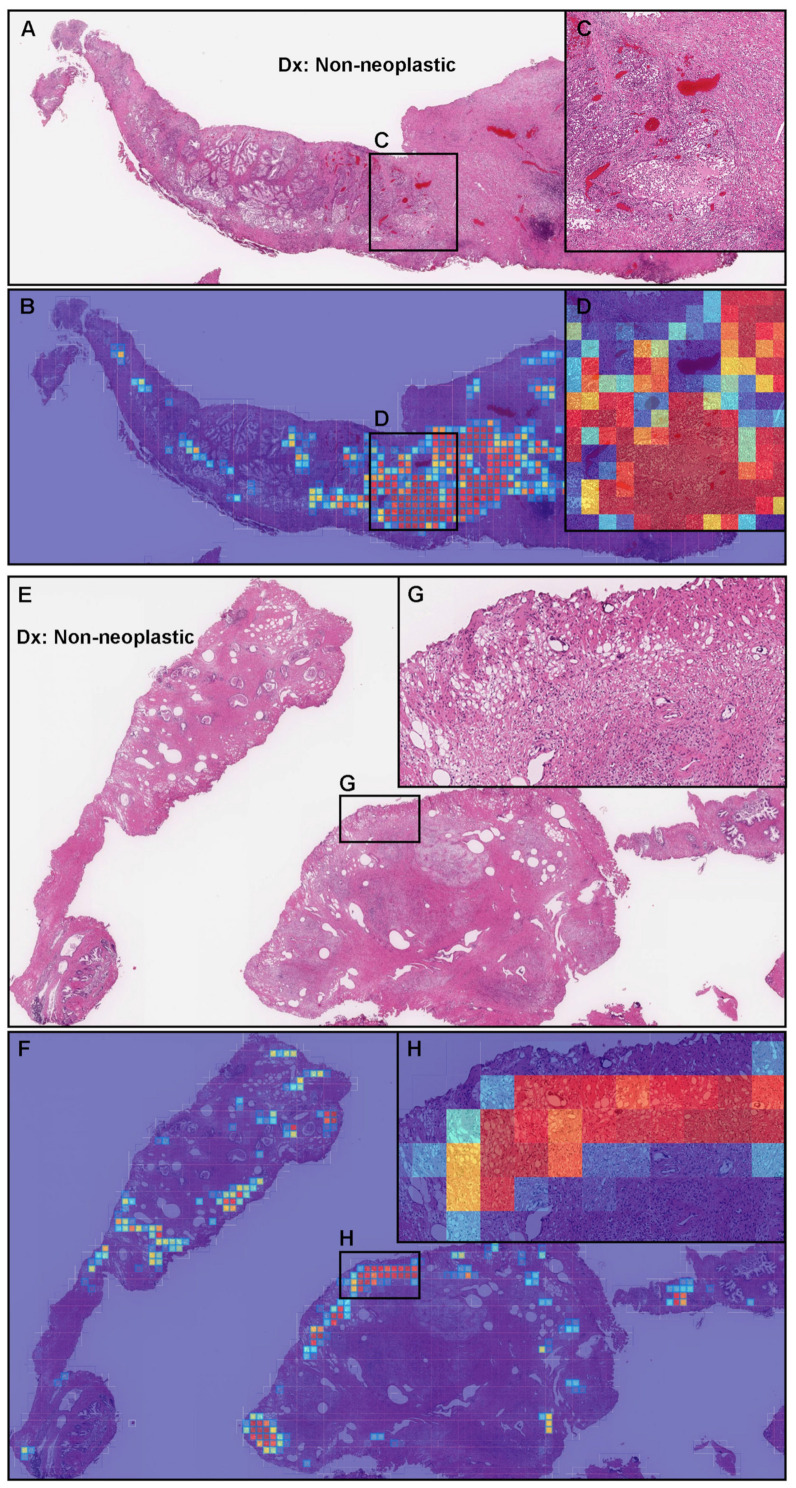
Representative examples of prostate adenocarcinoma false positive prediction outputs on cases from the transurethral resection of the prostate (TUR-P) test sets. Histopathologically, (**A**,**E**) are benign (non-neoplastic) lesions. The heat map images (**B**,**F**) exhibit false positive predictions of adenocarcinoma (**D**,**H**) using transfer learning from the colon poorly differentiated adenocarcinoma model (TL-colon poorly ADC-2 (20×, 512)). Inflammation with infiltration of inflammatory cells including foam cells (**C**) would be the primary cause of the false positives due to a morphology analogous to adenocarcinoma cells’ infiltration (**D**). The cauterized area of the marginal zone of the specimen (**G**) would be the primary cause of the false positives (**H**). The heat map uses the jet color map where blue indicates low probability and red indicates high probability.

**Table 1 diagnostics-12-00768-t001:** Distribution of the WSIs in the training and validation sets.

		Adenocarcinoma	Benign	Total
Training set	Hospital A	144	260	404
	Hospital B	100	75	175
	Hospital C	115	159	274
	Hospital D	56	118	174
	Hospital E	23	72	95
	Total	438	684	1122
Validation set	Hospital A	6	6	12
	Hospital B	6	6	12
	Hospital C	6	6	12
	Hospital D	6	6	12
	Hospital E	6	6	12
	Total	30	30	60

**Table 2 diagnostics-12-00768-t002:** Distribution of the WSIs in the test sets.

		Adenocarcinoma	Benign	Total
Biopsy	Hospitals A–C	250	250	500
	Hospital A	100	100	200
	Hospital B	100	100	200
	Hospital C	50	50	100
TUR-P	Hospitals A–B	162	1082	1244
	Hospital A	109	352	461
	Hospital B	53	730	783
Public dataset	TCGA	733	34	768

**Table 3 diagnostics-12-00768-t003:** ROC-AUC and log loss results for the various existing models on the prostate biopsy test sets.

Existing Models	ROC-AUC	Log Loss
Breast IDC (10×, 512)	0.704 [0.659–0.751]	0.947 [0.816–1.064]
Breast IDC, DCIS (10×, 224)	0.692 [0.644–0.735]	1.413 [1.282–1.566]
Colon ADC, AD (10×, 512)	0.553 [0.507–0.611]	1.525 [1.350–1.711]
Colon poorly ADC-1 (20×, 512)	0.795 [0.756–0.835]	0.572 [0.513–0.637]
Colon poorly ADC-2 (20×, 512)	0.817 [0.782–0.855]	0.522 [0.475–0.569]
Stomach ADC, AD (10×, 512)	0.706 [0.662–0.753]	1.391 [1.248–1.569]
Stomach poorly ADC (20×, 224)	0.724 [0.681–0.767]	0.598 [0.565–0.629]
Stomach SRCC (10×, 224)	0.804 [0.763–0.839]	0.998 [0.894–1.114]
Pancreas EUS-FNA ADC (10×, 224)	0.774 [0.735–0.817]	0.587 [0.544–0.629]
Lung carcinoma (10×, 512)	0.702 [0.659–0.751]	1.398 [1.2560–1.546]

**Table 4 diagnostics-12-00768-t004:** ROC-AUC and log loss results of the three different models for prostate adenocarcinoma on the biopsy, TUR-P, and TCGA test sets.

		TL-Colon Poorly ADC-2 (20×, 512)	
		ROC-AUC	Log-Loss
Biopsy	Hospitals A–C	0.967 [0.955–0.982]	0.288 [0.210–0.354]
	Hospital A	0.978 [0.966–0.995]	0.209 [0.117–0.276]
	Hospital B	0.972 [0.948–0.988]	0.378 [0.276–0.536]
	Hospital C	0.967 [0.922–0.993]	0.265 [0.117–0.512]
TUR-P	Hospitals A–B	0.845 [0.806–0.883]	4.152 [4.047–4.253]
	Hospital A	0.909 [0.865–0.947]	3.269 [3.089–3.451]
	Hospital B	0.737 [0.657–0.810]	4.672 [4.559–4.798]
Public dataset	TCGA	0.987 [0.977–0.995]	0.074 [0.055–0.095]
		**EfficientNetB1 (20**×**, 512)**	
		**ROC-AUC**	**Log-Loss**
Biopsy	Hospitals A–C	0.971 [0.955–0.982]	0.256 [0.188–0.349]
	Hospital A	0.979 [0.962–0.993]	0.209 [0.110–0.322]
	Hospital B	0.978 [0.963–0.992]	0.279 [0.167–0.398]
	Hospital C	0.977 [0.959–1.000]	0.306 [0.037–0.406]
TUR-P	Hospitals A–B	0.803 [0.765–0.848]	5.113 [4.976–5.252]
	Hospital A	0.875 [0.834–0.923]	4.308 [4.059–4.550]
	Hospital B	0.670 [0.597–0.753]	5.588 [5.411–5.729]
Public dataset	TCGA	0.945 [0.912–0.973]	0.101 [0.067–0.147]
		**EfficientNetB1 (10**×**, 224)**	
		**ROC-AUC**	**Log-Loss**
Biopsy	Hospitals A–C	0.739 [0.691–0.783]	0.631 [0.545–0.724]
	Hospital A	0.751 [0.668–0.810]	0.605 [0.511–0.744]
	Hospital B	0.929 [0.885–0.970]	0.335 [0.223–0.427]
	Hospital C	0.472 [0.348–0.572]	1.278 [0.979–1.501]
TUR-P	Hospitals A–B	0.804 [0.760–0.847]	0.392 [0.369–0.417]
	Hospital A	0.771 [0.705–0.820]	0.424 [0.384–0.474]
	Hospital B	0.928 [0.859–0.980]	0.373 [0.347–0.408]
Public dataset	TCGA	0.578 [0.497–0.661]	1.575 [1.481–1.657]

**Table 5 diagnostics-12-00768-t005:** Accuracy, sensitivity, specificity, and F1-score results of the transfer learning model (TL-colon poorly ADC-2 (20×, 512)) from the existing colon poorly differentiated adenocarcinoma model for prostate adenocarcinoma on the biopsy, TUR-P, and TCGA test sets.

		Accuracy	Sensitivity	Specificity	F1-Score
Biopsy	Hospitals A–C	0.918 [0.894–0.942]	0.912 [0.878–0.946]	0.924 [0.888–0.955]	0.918 [0.889–0.941]
	Hospital A	0.945 [0.920–0.980]	0.930 [0.897–0.989]	0.960 [0.915–0.991]	0.944 [0.920–0.981]
	Hospital B	0.925 [0.885–0.955]	0.890 [0.824–0.944]	0.960 [0.912–0.991]	0.922 [0.878–0.955]
	Hospital C	0.940 [0.880–0.980]	0.900 [0.796–0.964]	0.980 [0.921–1.000]	0.938 [0.870–0.978]
TUR-P	Hospitals A–B	0.894 [0.866–0.922]	0.700 [0.603–0.813]	0.926 [0.896–0.950]	0.618 [0.561–0.675]
	Hospital A	0.918 [0.889–0.939]	0.798 [0.712–0.867]	0.955 [0.930–0.975]	0.821 [0.749–0.871]
	Hospital B	0.890 [0.867–0.909]	0.415 [0.265–0.529]	0.925 [0.906–0.940]	0.339 [0.212–0.424]
Public dataset	TCGA	0.949 [0.934–0.965]	0.948 [0.932–0.965]	0.971 [0.906–1.000]	0.973 [0.964–0.981]

## Data Availability

The datasets generated during and/or analyzed during the current study are not publicly available due to specific institutional requirements governing privacy protection, but are available from the corresponding author upon reasonable request. The datasets that support the findings of this study are available from Shinyukuhashi, Wajiro, Shinkuki, Shinkomonji, and Shinmizumaki hospitals (Kamachi Group Hospitals, Fukuoka, Japan), but restrictions apply to the availability of these data, which were used under a data use agreement that was made according to the Ethical Guidelines for Medical and Health Research Involving Human Subjects as set by the Japanese Ministry of Health, Labour and Welfare, and so are not publicly available. However, the data are available from the authors upon reasonable request for private viewing and with permission from the corresponding medical institutions within the terms of the data use agreement and if compliant with the ethical and legal requirements as stipulated by the Japanese Ministry of Health, Labour and Welfare. The external prostate TCGA datasets are publicly available through the Genomic Data Commons (GDC) Data Portal (https://portal.gdc.cancer.gov/ accessed on 1 April 2020).

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
