# Peer review of "A Deep Learning Model for Prostate Adenocarcinoma Classification in Needle Biopsy Whole-Slide Images Using Transfer Learning"

_diagnostics, 2022, doi:10.3390/diagnostics12030768_

Round 1

Reviewer 1 Report

I am pleased to read your article. I've heard of the transfer learning technique, but I think it's an excellent way to apply it to pathology. It is also exciting that the previous training set and TCGA image to which Rasnet or image nets algorithm is used are all possible as training sets. It would be good if pictures such as false positives were included in the image appendix in the thesis.

Author Response

Reviewer 1

I am pleased to read your article. I've heard of the transfer learning technique, but I think it's an excellent way to apply it to pathology. It is also exciting that the previous training set and TCGA image to which Rasnet or image nets algorithm is used are all possible as training sets. It would be good if pictures such as false positives were included in the image appendix in the thesis.

Response: False-positive figures (biopsy and TUR-P) were already included in the manuscript. Figure 4 already shows a false positive case.

Reviewer 2 Report

  • The authors adequately presented methods but the results needs a bit more discussion other than just numbers. Please expand it.
  • Please align the figures and tables with the text. It gets a bit tiring, going back and forth from text and figures/tables.
  • Please provide a work flowchart diagram from data collection, pre-processing to algorithms.
  • Please compute the F1-score also, and mention accuracy, sensitivity, specificity and F1-score in the abstract.
  • Please mention why lateral cores increases cancer detection than central cores (lines 34-35).
  • Some word are repeated. Please check the text thoroughly and correct them. E.g. "the partial" in line 101.
  • Please add the possible reasons on why the TUR-P results are low.
  • Please read the following article, and add possible pitfalls in the deep learning models. Article: https://www.nature.com/articles/s42256-021-00307-0
  • The dataset used for training and test purposes and not balanced for all the cases. Can the authors please discuss how this might have affected the model outcomes? E.g. producing biased results.
  • Also, please provide results with atleast 20% data used for validation, and 80-90% data for training. According to the manuscript, less than 10% data are used for validating the models, and since the dataset are not balanced, this might cause serious biasing results.
  • Please add finals results from k cross validation only , with 80% data used for training.  This should provide a better performance on the overall dataset. Using a specific fixed split dataset might provide biased results.

Author Response

Reviewer 2

The authors adequately presented methods but the results needs a bit more discussion other than just numbers. Please expand it.

Response: The discussion is deferred to the discussion section. The results section was used to present the numerical results. We will check with the editor to see whether it’s fine to move some of the discussion to the results section; however, that might create less distinction between both.

Please align the figures and tables with the text. It gets a bit tiring, going back and forth from text and figures/tables.

Response:  We used the official provided latex template by the journal and it automatically aligns the figures and tables based on their size. As most of the figures are large; they tend to automatically be moved to the end of the page.

Please provide a work flowchart diagram from data collection, pre-processing to algorithms.

Response: Figure 1 already provides a flowchart for the pre-processing with slide tiling and the main training algorithm. There is not much to illustrate in terms of data collection as there was no annotation being carried out on the WSIs.

Please compute the F1-score also, and mention accuracy, sensitivity, specificity and F1-score in the abstract.

Response: We’ve added the F1-score in Table 1. To make the results consistent with the numerous publications in the field, most of which do not report the F1-score, we will only keep the report of the AUC in the abstract.

Please mention why lateral cores increases cancer detection than central cores (lines 34-35).

Response: That is simply based on the results from the referenced systematic review [1]. We have rewritten the sentence a bit to make it clear it is so.

[1] Eichler K, Hempel S, Wilby J, Myers L, Bachmann LM, Kleijnen J. Diagnostic value of systematic biopsy methods in the investigation of prostate cancer: a systematic review. J Urol. 2006 May;175(5):1605-12. doi: 10.1016/S0022-5347(05)00957-2. PMID: 16600713.

Some word are repeated. Please check the text thoroughly and correct them. E.g. "the partial" in line 101.

Response: Fixed. Thanks.

Please add the possible reasons on why the TUR-P results are low.

Response: This is potentially due to the presence of prostate hyperplasia which morphologically mimics prostate adenocarcinoma. We have added a further sentence in the discussion mentioning this.

Please read the following article, and add possible pitfalls in the deep learning models. Article: https://www.nature.com/articles/s42256-021-00307-0

Response: That paper deals with the pitfalls specific to the application of machine learning on CT images, and more specifically for COVID-19. With CT images, obtaining a high quality curated dataset is more difficult than in the case of histopathology.  What is mainly required in the histopathology case is more clinical validation studies of the proposed models. We have extended the last paragraph in the discussion to mention this.

The dataset used for training and test purposes and not balanced for all the cases. Can the authors please discuss how this might have affected the model outcomes? E.g. producing biased results.

Response: As referenced in table 2, the biopsy test sets are equally balanced between the two classes (benign and adenocarcinoma). During training, we used the commonly used approach of oversampling to ensure that the training batches were balanced between the two classes in order to mitigate the effect of the biases. The TUR-P and the TCGA set were not included in the training or validation set, only in the test set; and they were used as additional test sets to evaluate its generalization on images that are potentially out of distribution and not similar to the training set. As the model is a biopsy-based model, the main results are those on the biopsy test set, and given they are equally balanced, there is no bias.

Also, please provide results with atleast 20% data used for validation, and 80-90% data for training. According to the manuscript, less than 10% data are used for validating the models, and since the dataset are not balanced, this might cause serious biasing results.

Response: The validation set was equally balanced between adenocarcinoma and benign. In addition, the training set is roughly balanced (438 vs 684). We had also used further balancing on the batch level during training. Taking all this into account, and the fact the main test sets are equally balanced, there is no room for serious biasing of the results.

Please add finals results from k cross validation only , with 80% data used for training.  This should provide a better performance on the overall dataset. Using a specific fixed split dataset might provide biased results.

Response: The k cross validation is useful in cases when the dataset size is very small. And the aim of it is to get a better estimate of the performance of the model on an extremely small test set. This is not the case with this study. The WSIs are massive images, resulting in millions of tiles for training, and such a dataset is not small. There would be no further benefit from doing k cross validation. In addition, training the model on a single GPU took 6 days and it would take significantly longer to perform which is why k cross validation is not carried out in deep learning with large datasets. And again, our main biopsy test sets were fully balanced. There is no room for bias in that case.

Round 2

Reviewer 2 Report

I am satisfied with the author's response.